# Effects of Different Concentrations of Ganpu Tea on Fecal Microbiota and Short Chain Fatty Acids in Mice

**DOI:** 10.3390/nu13113715

**Published:** 2021-10-22

**Authors:** Chen Wang, Zhipeng Gao, Yujiao Qian, Xiang Li, Jieyi Wang, Jie Ma, Jiajing Guo, Fuhua Fu

**Affiliations:** 1Longping Branch, Graduate School of Hunan University, Changsha 410125, China; ch13750@163.com (C.W.); yujiaoqian2021@163.com (Y.Q.); fjrxloveyzby@outlook.com (X.L.); wangjieyi1996@163.com (J.W.); 2International Joint Lab on Fruits &Vegetables Processing, Quality and Safety, Hunan Key Lab of Fruits &Vegetables Storage, Processing, Quality and Safety, Hunan Academy of Sciences, Hunan Agriculture Product Processing Institute, Changsha 410125, China; 3College of Animal Science and Technology, Hunan Agricultural University, Changsha 410128, China; gaozhipeng627@163.com (Z.G.); jie_ma2020@yeah.net (J.M.)

**Keywords:** Ganpu tea, Pu-erh tea, tangerine peel, lipid metabolism, fecal microbiota modulatory, short-chain fatty acids

## Abstract

Ganpu tea is composed of tangerine peel and Pu-erh tea. Current research suggests that both products can interact with gut microbes and thus affect health. However, as a kind of compound health food, little information is available about the effect of Ganpu tea on intestinal microorganisms. In this study, the basic physiological parameters (body weight, white adipose tissue and serum fat), the regulation of intestinal microorganisms and content of short-chain fatty acids (SCFAs) in feces of healthy mice were studied. The Ganpu tea can reduce the weight gain of mice and the increase in white adipose tissue (*p* < 0.01). After the intake of Ganpu tea, the abundance of Bacteroidetes increased (*p* < 0.05), whereas that of Firmicutes decreased (*p* < 0.01), indicating the latent capacity of Ganpu tea in adjusting the gut microbiota. Moreover, Ganpu tea differentially affected the content of different types of SCFAs in feces. Ganpu tea at the lowest concentrations showed positive effects on the concentrations of SCFAs such as acetic acid and propionic acid, whereas the concentration of butyric acid was decreased. For branched short-chain fatty acids (BSCFAs) such as isobutyric acid, isovaleric acid, etc., Ganpu tea reduced their concentrations. Our results indicated that Ganpu tea may have positive effects on preventing obesity in humans, but further research is needed before introducing such dietary therapy.

## 1. Introduction

Pu-erh tea is a fully fermented tea that is often produced in Yunnan Province, China [1]. In daily life, Pu-erh tea is a popular healthy drink, and it is related to the lowering of blood lipid, anti-oxidation and hypoglycemia [2]. Tangerine peel is the dried peel of orange and is a cultivated variety in the rue family; it can not only be used as food but also as a traditional Chinese medicine [3]. Ganpu tea is a kind of compound tea that is made from Pu-erh tea and tangerine peel through baking, aging and other processes. Ganpu tea has the citrus fruit flavor and old flavor of Pu-erh tea. The production process of Ganpu tea removes the internal contents of the whole fresh green orange and leaves the outer orange peel; then, the peel is mixed and dried together with Pu-erh tea and stored in a cool and dry place [4]. Traditionally, the longer the tangerine peel and Pu-erh tea are stored, the better their quality will be [5,6].

Gut microbes can respond to food intake within a short period of time and play a principal function in human nutrition and health [7,8]. The intestinal microbiota reciprocates the multifarious perception and signal path of the host, thereby regulating the activities of the endocrinium, immunologic system and systema nervosum, and is closely related to metabolic diseases [9], such as obesity [10], diabetes [11,12] and inflammation [13]. Diet is an important influencing factor in the interaction between the gut microbiome and the host and the consequent influence on host health and disease [14]. Pu-erh tea and tangerine peel contain abundant polyphenols, such as flavonoids and tea polyphenols [15,16], which can enhance the beneficial content of the intestinal microbiota and reduce harmful bacteria to improve the body’s healthy. After oral intake of the tea, the active components of polyphenols in Pu-erh tea [17] and tangerine peel [18] can regulate the composition of intestinal microorganisms and the production of short-chain fatty acids (SCFAs). The interaction between the tea and gut microbes may partly explain the tea’s health benefits. As a kind of health drink, Ganpu tea appears in people’s daily life. Studies should focus on how the intestinal microbiota and SCFAs change after drinking Ganpu tea, and whether they can play a beneficial regulatory role on the host. This is of great significance for the application of Ganpu tea as dietary therapy.

This study aimed to use C57 mice as model animals to investigate the effects of different concentrations of Ganpu tea on lipid metabolism, and the regulation of fecal microbiota and SCFAs. Ganpu tea may have beneficial effects on lipid metabolism and obesity prevention in humans by modulating the composition and abundance of gut microbes and the production of SCFAs, such as upregulating the proportion of microbiota that are metabolically beneficial and the SCFA content. Then, the body weight, relative weight of adipose tissue and serum indexes of the mice were measured. Hematoxylin and eosin (H&E) staining and section analyses were performed on fat and organ tissues. In addition, the intestinal microbiome was analyzed by high-throughput 16 SrRNA gene sequencing. Finally, the condition of SCFAs was determined. In order to evaluate and compare microbial activity levels between experimental groups, the total SCFA concentration of each sample was calculated. The findings of this study may provide a theoretical basis for the health care effect of Ganpu tea and its further use as dietary therapy.

## 2. Materials and Methods

### 2.1. Sample Preparation

The Ganpu tea sample with a storage time of four years used in this experiment was provided by Chakeng Village, Xinhui (Guangzhou, China). After separating the orange peel and Pu-erh tea, the mass ratio of tea and peel was about 8:2 after weighing calculation. After the Ganpu tea samples were crushed, 40 g Ganpu tea (8 g orange peel, 32 g Pu-erh tea) were extracted with boiling water 3 times, with extraction volume ratios of 1:20, 1:20 and 1:10, respectively, and the extraction times were 20 min, 20 min and 10 min. The water extract of Ganpu tea with a concentration of 0.4 g/mL was obtained by combining the samples after three times of extraction and evaporation concentration.

### 2.2. Experimental Design and Diets

Forty C57BL/6J female mice (4 weeks old,13 g ± 3 g) were purchased from tianqin Biotechnology Co., Ltd (Changsha, Hunan) and fed in the specific pathogen-free (SPF) requirement. C57 mice were used because the mouse strain has completed genome sequencing and is not prone to many tumors, but can produce diet-induced obesity and atherosclerosis. Therefore, it is often used as an obesity model, a type II diabetes model, an atherosclerosis model and a related research model for physiology and pathology. It is the most widely used strain in animal testing currently. The strain can guarantee high stability in genetic background and consistency of experimental data. All the experimental processes were carried out in the animal experimental base of the Hunan Agricultural University. Moreover, the animal model and experimental procedures used in this experiment were approved by the Hunan Agricultural University Institutional Animal Care and Use Committee (202005). The mice were kept in rooms with temperatures ranging from 20 to 23 °C, relative humidity ranging from 50 to 65% and alternating light for 12 h. Mice were given adaptive feeding in their new environment for a week before the experiment began. Mouse maintenance feed was purchased from slake Jingda company (Changsha, Hunan). The main raw materials were as follows: Soybean meal, fish meal and brewer’s yeast powder were used as a protein source; vegetable oil was used as a fat source; corn and wheat were used as a carbohydrate source; alfalfa and perennial grass powder and wheat bran were used as a fiber source; vitamins and minerals were added in addition. The nutritional composition was as follows: Crude protein > 18%, ether extract > 4%, crude fiber < 5%, crude ash powder < 8%, water content < 10%, calcium 1.0–1.8%, phosphorus 0.6–1.2%.

After adaptive feeding, forty mice were randomly divided into four groups (*n* = 10/group, two animals per cage, cage model: MJ1): Control group (CON), low concentration Ganpu tea (GPTL) group (0.1 g/mL, 0.2 mL/d), medium concentration Ganpu tea (GPTM) group (0.2 g/mL, 0.2 mL/d), and high concentration Ganpu tea (GPTH) group (0.4 g/mL, 0.2 mL/d). The experimental mice were gavaged with different concentrations of Ganpu tea water extract every day for 3 consecutive weeks. Meanwhile, the control group was given the same volume of normal saline. Gavage is a common method of drug administration in animal experiments. In the experiment, the animals were immobilized, administered with a gavage needle and the liquid was injected directly from the animal’s mouth into its stomach.

### 2.3. Sample Collection and Basic Index Determination

The body weight and feed intake of the mice were weighed weekly and recorded. Fecal samples were collected the day before the mice were killed. Individual mice were placed in clean and sterile mouse cages. Fecal samples were collected externally by natural defecation, filled in sterile centrifuge tubes and frozen in liquid nitrogen. Two to three feces samples were collected separately for each mouse as individual samples. After the mice were anesthetized with ether, the eyeballs of the mice were removed and the retro-orbital blood was collected. After blood collection, mice were killed by cervical dislocation. Adipose tissue was collected and weighed immediately, then stored in the fixative for sectioning analysis. Blood samples were collected and coagulated, then centrifuged at 4 °C and 5000 rpm for 10 min to obtain serum samples. All samples were collected and stored at −80 °C until the next step of analysis. The randomized controlled trials registration number is: PRJNA729594.

### 2.4. Serum Biochemical Indicators Analysis

Serum samples obtained after centrifugation were analyzed with an automatic biochemical analyzer (KHB, Excellence 450, Shanghai, China) for serum indicators, including serum triglyceride (TG), total serum cholesterol (TC), total high-density lipoprotein cholesterol (HDL-C), and total low-density lipoprotein cholesterol (LDL-C).

### 2.5. Histological Analyses

The tissue sections were made and stained to observe the tissue morphology. First, the paraffin section was dewaxed to water: The paraffin section was put into xylene, anhydrous ethanol, 75% alcohol in turn and then washed with tap water. Then the slices were dyed with hematoxylin staining solution for 3–5 min. After washing, the differentiation solution was differentiated, and then washed with water. The blue returning solution was washed with running water. Then eosin staining was carried out: The slices were dehydrated with 85% and 95% gradient alcohol for 5 min, and then stained with eosin staining solution for 5 min. The next step was dehydration sealing: The slices were put into anhydrous ethanol and xylene in turn and then sealed with neutral gum. Finally, microscope examination, image acquisition (Pannoramic MIDI, Budapest, Magyarország) and analysis were carried out.

### 2.6. DNA Extraction and 16S rRNA Sequencing

Total bacterial DNA was extracted from the fecal samples. The 16sRNA gene was amplified by PCR using primers and sequenced on the platform. The V3-V4 hypervariable regions of the bacteria 16S rRNA gene were amplified with primers 338F (5′-ACTCCTACGGGAGGCAGCAG-3′) and 806R (5′-GGACTACHVGGGTWTCTAAT-3′) by a thermocycler PCR system (ABI GeneAmp^®^ 9700, Waltham, MA, USA). The resultant PCR products were extracted from a 1% agarose gel, then purified using an AxyPrep DNA Gel Extraction Kit (Axygen Biosciences, Union City, CA, USA) and further quantified using QuantiFluorTM-ST (Promega, Madison, WI, USA). Afterwards, the purified amplicons were pooled in equimolar and paired end sequenced (2 × 300) on an Illumina MiSeq platform (Illumina, San Diego, CA, USA) by following the standard protocols of Majorbio Bio-Pharm Technology Co., Ltd. The raw paired-end reads were merged using FLASH (version 1.2.11, Baltimore, MD, USA). All quality filtered sequencing reads were then clustered into operational taxonomic units (OTUs) based on a 97% sequence similarity according to Uparse (version 7.0, Tiburon, CA, USA).

### 2.7. Calculation of Diversity and Richness Index

The community richness and diversity were estimated by using Chao, ACE, Shannon index and Simpson index. The principal component analysis (PCA), non-metric multi-dimensional scale (NMDS) and principal cylindrical coordinate analysis (PCoA) at the phylum level were used to analyze the changes in the gut microbiota structure to estimate the extent to which the gut microbiota of the GPTL, GPTM and GPTH groups differed from that of the CON group. Chao is used to estimate the number of OTUs in a sample by the Chao 1 algorithm, which is often used to estimate the total number of species (https://mothur.org/wiki/chao/, 25 April 2021); ACE is also an index to estimate the total number of species. The algorithm is different from Chao 1 (https://mothur.org/wiki/ace/, 25 April 2021); Shannon is an index used to estimate microbial diversity in a sample. It is often used to reflect the alpha diversity of a community. The higher the Shannon value, the higher the community diversity (https://mothur.org/wiki/shannon/, 25 April 2021); Simpson is used to estimate the index of microbial diversity in samples, which is often used to quantitatively describe the biodiversity of a region. The Simpson index was negatively correlated with community diversity (https://mothur.org/wiki/simpson/, 25 April 2021). All of these analyses were carried out on the Majorbio Cloud platform (www.majorbio.com, 25 April 2021).

### 2.8. Fecal SCFA Analysis

Fecal samples were collected and processed before short chain fatty acid determination. After accurate weighing, 100 mg of sample was put into a 2 ml grinding tube; a grinding bead was added, as well as 1 ml of water (containing 0.5% phosphoric acid and 50 μg/ml internal standard 2-ethylbutyric acid); the sample was ground twice at 50 Hz for 3 min. Then an ice water bath was used for ultrasonic treatment for 30 min; 4 °C for 30 min; centrifugation at 4 °C 13,000× *g* for 15 min. All the supernatants were put into a new 1.5 ml centrifuge tube. Finally, 500 μL ethyl acetate was extracted from the supernatant solution, vortex mixing took place, followed by ice water bath ultrasound for 10 min, 4 °C 13,000× *g* centrifugation for 10 min, and the supernatant solution was analyzed on the machine. The fecal SCFAs were tested by using Agilent 8890 b-5977b GC/MCD GCMS (Agilent Technologies Inc. CA, USA) and the analysis of SCFAs was carried out as previously described [19]. The chromatographic conditions were as follows: HP FFAP capillary column (30 m × 0.25 mm × 0.25 μM, Agilent J&W Scientific, Folsom, CA, USA); carrier gas of high purity helium (purity not less than 99.999%); flow rate of 1.0 ml/min, injection temperature of 260 °C; injection volume 1 μL; split injection, split ratio 10:1; solvent delay 2.5 min. The initial temperature of the column incubator was 80 °C, the temperature was programmed to 120 °C at 40 °C/min, the temperature was programmed to 200 °C at 10 °C/min and then it ran at 230 °C for 3 min. The temperature of the ion source was 230 °C, the temperature of the quadrupole was 150 °C, the temperature of the transmission line was 230 °C and the electron energy was 70 ev. The scanning mode was SIM.

The default parameters of mashunker (Agilent, v10.0.707.0) were used to automatically identify and integrate the ion fragments of the target SCFAs and to assist manual inspection. The actual content of SCFAs in the sample was calculated by calculating the detection concentration of the sample through the standard curve (Appendix A). Appendix A shows the gas chromatographic diagram of the standard solution and sample according to their peak time. The individual SCFA content (μg/mg) was calculated according to the following formula:(1)$$W\left(\mu g/mg\right)=\frac{C\;\times \;V\;\times \;N}{m}$$
where *C* refers to the concentration of SCFA in fresh fecal sample solution (μg/mL); *V* represents the constant volume (mL); *N* depicts the dilution factor; and m represents the weight of the sample (mg).

### 2.9. Statistical Analysis

All the data were expressed as the mean ± standard deviation (SD). Statistical significance between groups was evaluated by one-way ANOVA, followed by Tukey’s test using SPSS Statistics 19.0 (IBM, Chicago, IL, USA). The GraphPad Prism 8.0 (GraphPad Software, San Diego, CA, USA) was used for data visualization. The Kruskal–Wallis rank sum test was used to compare the Firmicutes and Bacteroidetes in the four groups. * *p* < 0.05 and ** *p* < 0.01 were considered significant or extremely significant.

## 3. Results

### 3.1. Basic Indicators

In the determination of the effects of different concentrations of Ganpu tea on healthy mice, feed intake and body weight were recorded (Figure 1). Compared with the CON group, the GPTL (fed for 14 days) showed a similar feed intake trend (*p* > 0.05). Feed intake decreased first and then increased. Then, the values of the GPTM and GPTH groups in the second week were significantly higher than the CON groups (*p* < 0.05, Figure 1A). At the beginning of the experiment, the four groups had similar body weights. After feeding for three weeks, the weight of the GPT groups was smaller than the CON group. The higher the sample concentration, the smaller the weight of the GPT groups (Figure 1B). After feeding for three weeks, the feed intake of the GPTM and GPTH groups was higher than the CON and GPTL groups. However, the weight of the CON and GPTL groups was higher than the GPTM and GPTH groups.

Adipose tissue is a momentous apparatus for the development of obesity. To further investigate the result of Ganpu tea intake on fat growth in mice, the subcutaneous adipose tissue (SAT), abdominal adipose tissue (AAT), perirenal adipose tissue (PEAT) and small-intestine (SI) of mice were collected, weighed and compared to the body weight (Figure 1). As shown in Figure 1(A), the CON group had a significantly higher relative weight of SAT to body weight than the GPTM group (*p* < 0.01) but showed no significant difference with the other GPT groups (*p* > 0.05). The CON group’s AAT was significantly higher than those of the GPTL, GPTM (*p* < 0.01) and GPTH groups (*p* < 0.05) (Figure 1D). The PEAT showed similar trends to the SAT. The SI showed an opposite trend to the adipose tissue. The GPTH group showed a significantly higher SI than the CON group (*p* < 0.001), and significant differences were observed between the GPTL and GPTH groups (*p* < 0.01).

### 3.2. White Adipose Tissue Steatosis in the Study Animals

In this study, we used hematoxylin and eosin (H&E) staining on abdominal adipose tissue (AAT) to evaluate the adipocyte size. The histopathological changes of the main AAT in mice are shown in Figure 2. H&E staining analysis of AAT showed that mice treated with Ganpu tea exhibited smaller cell size than the CON group, and the GPTH group had a more obvious inhibitory effect on adipocyte enlargement. Under the same magnification ratio (400×), the number of adipocytes in the GPT groups was significantly higher than that of the CON group (Figure 2).

### 3.3. Serum Lipid Indices

Serum high-density lipoprotein (HDL) and low-density lipoprotein (LDL) in the CON group were significantly higher than the GPTL group (*p* < 0.05). The total cholesterol (TC) and triglycerides (TG) showed no significant differences among the four groups (Figure 3).

### 3.4. Overall Information of Fecal Microbiota

To explore the effect of Ganpu tea on the gut microbiota of mice, we sequenced the V4 super variable area of 16S rDNA on a high-throughput sequence platform. Through sequencing analysis, 2,258,760 effective sequences were obtained, the number of effective bases was 949,004,677 and the average length of sequences was 420. All valid reads were aggregated into operational taxonomic units (OTUs) based on the 97% sequence similarity level. All 851 OTUs were detected. In addition, 584 OTUs were common to all groups. A total of 9, 8, 11 and 28 OTUs were unique to the CON, GPTL, GPTM and GPTH groups, respectively.

### 3.5. Gut Microbiota Diversity and Richness

Dietary interventions had significant effects on the diversity of intestinal microbes, including alpha diversity [20]. In terms of the ACE and Chao indexes, a significant difference was observed in the microbial richness between the CON, GPTH, GPTL, and GPTH groups (Figure 4A,B). In addition, the microbial richness of all the GPT groups was higher than that of the CON group. However, the richness of the GPTL and GPTM groups was not significantly different from the CON group. The microbial richness in feces was positively correlated with the concentration of samples. The microbial diversity of CON was significantly different from that of GPTH, as measured by the Shannon index. Furthermore, the microbial diversity of the GPTL group was significantly different from those of the GPTM and GPTH groups (Figure 4C,D).

The principal component analysis, non-metric multi-dimensional scale and principal cylindrical coordinate analysis (PCoA) at the phylum level were used to analyze the changes in the gut microbiota structure (Figure 5) to estimate the extent to which the gut microbiota of GPTL, GPTM and GPTH groups differed from that of the CON group. The PCoA showed that the microbial community in the GPT groups changed. Compared with the GPTM and GPTH groups, the CON and GPTL groups had more overlapping samples. The PCoA plot revealed that the CON and GPT groups were separated along the PC1 axis, which indicated that the intestinal microbiome structures of the experimental and control groups were significantly different.

Gut microorganisms were analyzed from different microbial composition and structure classification levels to explore the changes at different levels. At the phylum level, the top four microorganisms were Bacteroidetes, Firmicutes, Campilobacterota and Actinobacteriota (Figure 6A). In the GPT groups, the relative abundance of Firmicutes decreased, whereas that of Bacteroides increased; the Firmicutes/Bacteroidetes (F/B) ratio in the GPT groups decreased compared with that of the CON group. An increase in the number of Firmicutes is thought to improve the host’s ability to obtain energy from the diet, absorb calories and gain weight more effectively [21]. In this study, weight gain in mice was accompanied by a decrease in Firmicutes abundance. *norank_ f_ Muribaculaceae*, *Lactobacillus*, *Lachnospiraceae_ NK4A136_ group*, *Bacteroides* and *Helicobacter* were the supreme affluent microbial groups at the genus level (Figure 6B).

After the gavage of mice with Ganpu tea, six bacteria groups showed significant difference at the phylum level, including Bacteroidota, Firmicutes, Actinobacteriota, Verrucomicrobiota, Proteobacteria and Cyanobacteria (Figure 6C). As shown by the plot, the level of Firmicutes in the GPTM group was evidently lower than that of the CON group (*p* < 0.01), whereas the GPTH group showed the same trend (*p* < 0.001) (Figure 6C). Compared with the CON group, Bacteroidota in the GPT groups showed a significant increase. At the genus level, the species with significant differences between groups are *norank_f_Muribaculaceae*, *Lactobacillus*, *Dubosiella*, *Akkermansia*, *Turicibacter*, *Coriobacteriaceae_UCG-002* (Figure 6D).

Through LEfSe analysis, microbial groups with significant differences between groups (from phylum level to genus level) were shown in Figure 7. At the genus level, the CON group has four enriched genera, which are *Lactococcus*, *Gemella*, *Corynebacterium* and *Arthromitus*. Two enriched genera *Dubosiella* and *Bacilli* were observed in the GPTL group. There are six enrichment genera in the GPTM group: *Akkermansia*, *Coriobacteriaceae_UCG-002*, *Bifidobacterium*, *Faecalibaculum*, *Escherichia-Shigella* and *Rhodospirillales*. *Muribaculaceae*, *Turicibacter* and *Flavobacteriaceae* were the three enrichment genera in the GPTH group.

### 3.6. SCFAs

Dietary fiber is fermented by gut microbiota to engender SCFAs, which reduce the increase in fat content by reducing fat accumulation in adipose tissues and expediting energy wastage [22]. Figure 8 shows the SCFAs in the mouse fecal samples, including acetic acid, propionic acid, isobutyric acid, butyric acid, isovaleric acid, valeric acid, isohexanoic acid and hexanoic acid. In this study, acetic acid accounted for the highest proportion of SCFAs detected in feces of mice in the GPT groups, followed by propionic acid, butyric acid and valeric acid. However, butyric acid concentration was higher than propionic acid concentration in the CON group, which means that Ganpu tea inverted the propionic/butyric acid ratio. The change of acetic acid content was dose-dependent. The acetic acid content of the GPTL group was the highest, and was significantly higher than the GPTH group (*p* < 0.05). However, the acetic acid content of the GPTM and GPTH groups was lower than the CON group. The content of propionic acid in the GPTL group was significantly higher than the CON group and the GPTH group (*p* < 0.01), and higher than the GPTM group.

Compared with the CON group, the contents of isobutyric acid, valeric acid and isovaleric acid in GPT groups decreased. In addition, the concentration of hexanoic acid in the GPTM group was significantly lower than the CON group (*p* < 0.05). The concentration of isohexanoic acid in the GPT groups decreased significantly (*p* < 0.05, *p* < 0.01 or *p* < 0.001). The content of the four kinds of SCFAs in the GPT groups was negatively correlated with the concentration of Ganpu tea. A decrease in SCFA production means lowering of microbial activity in the GPT groups. The concentration of total SCFAs in the GPTL group was significantly higher than the GPTH group (*p* < 0.05), and higher than the CON and GPTM groups, but the total SCFAs in the GPTM and GPTH groups was lower than the CON group (Figure 7I).

The Spearman correlation between fecal microorganisms (genus level) and SCFAs was explored in 40 individuals in all groups (Figure 9). Family norank_f_ Peptococcaceae, norank _f_Lachnospiraceae and norank_f_Ruminococcaceae were positively correlated with acetic acid (*p* < 0.05), while genus *Parasutterella* was negatively correlated (*p* < 0.05). Another high content of SCFA propionic acid was negatively correlated with genus *Bacteroides* (*p* < 0.05). For butyric acid, family norank_f_Peptococcaceae, genus *Lactobacillus* and *Helicobacter* were positively correlated with it (*p* < 0.05), while family norank_f_Muribaculaceae, norank_rank_f_norank_o_Gastranaerophilales and genus *Ruminococcus*, *Parasutterella* and *Alloprevotella* were negatively correlated (*p* < 0.05).

## 4. Discussion

When food enters the gut, it interacts with the gut microbiota, causing changes in the microbiota composition of the gut. In turn, the changes in the bacterial community will have certain negative and positive effects on the body [23]. Obesity and several chronic metabolic diseases are caused by an imbalance in the bacterial community, and an increase in beneficial bacterial community will lead to the healthy development of the body. Ganpu tea is a special compound product composed of tangerine peel and Pu-erh tea. Both have beneficial effects on the intestinal microbiota of the human body [24,25]. This study aimed to explore the effects of Ganpu tea on the intestinal microbiota and SCFAs in normal-diet mice.

In this study, we observed that the administration of different concentrations of Ganpu tea for three weeks was not effective in reducing body weight compared with the normal saline groups. However, as the concentration rose, the weight showed a decline. In previous studies, continuous intake of Pu-erh tea resulted in significant weight loss in mice on a high-fat or normal diet [26]. The main reasons may be as follows: C57 mice were not prone to obesity and did not gain significant weight. In addition, in this study, mice were fed a standard mouse maintenance diet and, although Ganpu tea water extract was associated with weight loss, the effect was not significant in non-obese subjects. Compared with the CON group, the GPTM and GPTH groups consumed more food, while the GPTL group consumed less food. However, the weight gain of the GPT groups were less than the CON group, which also indicates that the water extract of Ganpu tea had a certain effect on reducing body weight gain. This suggests that the reduction in weight gain did not depend on food intake, but may be linked to the resistance to diet-induced obesity caused by differences in the composition and function of intestinal microorganisms [27].

White adipose tissue, which is composed of SAT and visceral adipose tissue, is closely related to glucose and lipid metabolism [28]. Pu-erh tea extract can significantly reduce the body weight and SAT gain of mice on a high-fat diet [29]. Ganpu tea can reduce adipose obesity by inhibiting the formation of adipose tissue and reducing fat mass [30]. This finding is consistent with the results of our study. In this study, the WAT of the GPT groups was remarkably lower than that of the CON group. The SAT of GPTM was evidently less than that of the CON group. While no significant difference was observed in the body weight between CON and GPT, a significant reduction in WAT was recorded. The feeding cycle of this experiment was three weeks. If the test cycle was prolonged, then the difference in white fat eventually leads to differences in body weight.

The small intestine is the main organ of nutrient absorption [31]. The weight of the small intestine relative to the body can reflect the absorption of the small intestine to a certain extent. This condition is related to the villi and wall thickness of the small intestine. Numerous absorbent cells are present in the epithelium of the small intestine [32]. The villi in the small intestine increase the superficies area to facilitate efficient absorption. The relative weight of the small intestine in the GPT groups was higher than that in the CON group, and it was positively correlated with the concentration of Ganpu tea. The GPTH group was significantly higher than the CON group and low-concentration groups. In the GPT groups of different concentrations, significant differences were observed between the GPTL and GPTH groups, which indicated that the intake of different concentrations of Ganpu tea had varied effects on the changes in the intestinal structure.

Obesity in white adipose tissue is characterized by an increased cell volume [33]. A relatively healthy state occurs when cells remain small but increase in number [34]. The AAT cells in the CON group were considerably larger than those in the GPT groups. Under the same amplification ratio, the fat cells in the GPT groups were smaller and more numerous, which indicated that the intake of Ganpu tea could keep the adipose tissue in a healthier state and prevent the adipose cell hypertrophy caused by diet, which can lead to obesity.

The environment has a greater influence on the composition of gut microbes than genetic diathesis. Diet has a significant influence on the composition and function of the gut microbiome in the human body [35]. Intestinal microbiota regulates various physiological processes and influences different host functions. The human gut contains about 100 trillion microorganisms, which have a huge effect on host physiology and homeostasis of cells [36]. Dietary interventions had significant effects on the diversity of intestinal microbes, including alpha diversity [20]. Alpha diversity can reflect the species richness and diversity of individual samples. The richness of species and community uniformity in sample communities have an influence on alpha diversity. Under the condition of the same species richness, greater community uniformity is considered as greater diversity. A high Shannon index or a small Simpson index indicated a high diversity of species in the sample. This index can reflect whether the results of sequencing are representative of the real conditions of microorganisms in the sample [37]. Diet can change the richness and diversity of a microbial community [38]. These changes are also reflected in our results. As measured by Chao richness and Shannon diversity index, the highest microbial diversity was noted in the GPTH group, followed by the GPTM and GPTL groups. By contrast, the CON group had the lowest microbial diversity. The intake of Ganpu tea can increase the richness and diversity of intestinal microbiota. Thus, it can maintain the dynamic balance of intestinal microbiota and has a certain role in preventing obesity.

Diet is not only a factor affecting the dynamic balance of an intestinal microbial population but also a superficies diathesis influencing the constitution of microorganisms [39]. In this study, PCoA at the phylum level showed that the intake of Ganpu tea significantly transformed the overall structure and composition of the gut microbiota. Gut microbiota is an additional contributing factor to the pathophysiology of obesity [40]. The main phyla of the human gut microbiota are Firmicutes, Bacteroidetes, Proteobacteria and Actinobacteria. Bacteroidetes is often linked to the development of obesity, and obesity is related to the relative abundance of Bacteroidetes and Firmicutes [40,41]. At the phylum level, compared with the CON group, the proportion of Bacteroidetes increased in GPT groups, whereas that of Firmicutes decreased. Firmicutes and Bacteroidetes play a role in the regulation of host energy homeostasis, and a high proportion of Firmicutes and a low proportion of Bacteroidetes may lead to obesity [42]. Experiments have proven that the F/B ratio is positively correlated with obesity [43]. Our data show that the body weight and relative fat weight of the CON group with high F/B ratio were greater than the GPT groups, which is consistent with this conclusion. The changes in these two phyla of bacteria also indicated that Ganpu tea may have a role in preventing obesity. In an existing study, Actinobacteria are pro-inflammatory factors and show a negative correlation with the serum TC level [44]. The number of Actinobacteria in the GPTH group was lower than the CON group, but the GPTL and GPTM groups was higher than the CON group. Moreover, Actinobacteria were enriched in the GPTM group. This regulation shows that Ganpu tea can increase the abundance of Actinobacteria in a certain concentration range. Ganpu tea has the opposite effect when it exceeds a certain concentration. The abundance of *norank_f_Muribaculaceae* increased significantly with the concentration of Ganpu tea, which may be related to the resistance to HFD [27].At the genus level, *Lactobacillus* has an anti-obesity effect and is negatively correlated with the TC level [45]. However, in this experiment, *Lactobacillus* decreased in the GPT groups compared with the CON group. This may be related to that *Lactobacillus* is associated with obesity and is usually enriched in obese mice [46].

Unabsorbed/digested food components in the small intestine is fermented by gut microbiota to engender SCFAs [47,48]. The formation of SCFAs is influenced by a variety of factors, including food intake, antibiotic treatment, and microbial population size [49,50]. At present, a large number of studies have proved that SCFAs play a key role in the interaction between intestinal microbial activity and host metabolism [51]. The principal SCFAs in feces are acetic acid, propionic acid and butyric acid, which have been studied in detail and considered beneficial [47]. The main source of SCFA is carbohydrates, but the amino acids valine and leucine obtained from protein decomposition can be transformed into isobutyric acid and isovaleric acid, called branched chain SCFA (BSCFA), which makes little contribution to the total SCFAs yield. BSCFAs are not well characterized, but are generally considered to be harmful to the intestine [52]. BSCFAs (isobutyric acid and isovaleric acid) in the GPT groups decreased in varying degrees, which also proved the beneficial effect of Ganpu tea on intestinal health.

SCFAs can reduce the increase in fat content by reducing fat accumulation in adipose tissues and expediting energy wastage [22]. The abundance of Firmicutes and Bacteroidetes is linked to the level of propionic acid in fecal samples [53]. Intestinal microbiota may participate in a host signal transduction mechanism through microbial metabolites and use SCFAs as energy substances to provide nutrition for host cells. SCFAs may also regulate the occurrence of metabolic syndrome by regulating glucose and lipid metabolism [54,55]. Butyric acid and acetic acid can prevent diet induced obesity without loss of appetite, while propionic acid can reduce food intake, and butyric acid, propionic acid and acetic acid can prevent diet induced obesity and insulin resistance [56]. The concentration of propionic acid in the GPTL group was higher than the CON group. This means that the lowest concentration of Ganpu tea has the potential to prevent diet induced obesity. This is consistent with the result that GPTL group had the least feed intake. Acetic acid may be a function related metabolite because it promotes anti lipolytic activity through GPR43(a free fatty acid receptor) in WAT [57]. Acetic -dependent GPR43 stimulation in the WAT also improved glucose and lipid metabolism [56]. These data suggest that acetic acid may be metabolically beneficial through GPR43 activation in WAT. Then, they may explain that the relative weight of WAT in GPTL group is smaller than that in the CON group. The production of acetic acid and propionic acid is related to Bacteroides [58]. The concentrations of acetic acid and propionic acid increased with the abundance of Bacteroides in the GPTL group. Muribaculaceae was increased in abundance in the GPT groups, and the fermentation end product of this bacterium was propionic acid, while the concentration of propionic acid was greater in the GPT groups than in the CON group, which also corresponded with the previous experimental results [59]. *Akkermansia* is enriched in the GPTM group and has the ability to produce acetic acid and propionic acid [60]. It is negatively correlated with metabolic disorders in some clinical studies, and can protect mice from diet-induced obesity [61]. In the correlation analysis, *Akkermansia* was positively correlated with acetic acid and propionic acid, which was consistent with previous studies. The abundance of *Akkermansia* in the GPT groups was significantly higher than the CON group, indicating that Ganpu tea may avoid obesity caused by diet by increasing the abundance of *Akkermansia.*

However, for butyric acid, the concentration of the GPT groups decreased compared with the CON group. Studies have shown that supplementing lactic acid produced by lactobacillus can promote the growth of butyric acid production related bacteria and increase the accumulation of butyric acid [62]. Therefore, the concentration accumulation of *Lactobacillus* and butyric acid is in direct proportion, which is consistent with the analysis in Figure 9. The abundance of *Lactobacillus* in GPT groups is lower than that in the CON group, which may be one of the reasons for the decrease of butyric acid concentration in the GPT group. This phenomenon was also observed in previous experiments. Salazar et al. reported that the levels of acetic acid, propionic acid and caproic acid were significantly lower in obese individuals given 16 g inulin fructose than those given 16 g maltodextrin and the experimenter detected a greater abundance of bacteria in the intervention group [63]. This means that the relationship between the abundance of intestinal microorganisms and the concentration of short-chain fatty acids is complex. A limitation for SCFA and BSCFA production assays is that the examined samples for this experiment were mouse feces. The results reflect concentrations at the end of the digestion, but not necessarily those in other parts of the colon. In addition, the concentration effect of the absorption process for short-chain fatty acids was also considered.

## 5. Conclusions

This study explored the 21-day intervention effects of Ganpu tea on the lipid metabolism, intestinal microbiota and SCFAs in mice. Our study showed that Ganpu tea can reduce the growth of body weight. For the lipid metabolism, Ganpu tea can inhibit the growth of adipose tissue by reducing the cell size of white adipose tissue. The abundance of Bacteroidetes increased, whereas that of Firmicutes decreased. The change trend of fecal microbiota shows that Ganpu tea has the potential ability to regulate the host intestinal microbiota, and is beneficial to human health. Moreover, Ganpu tea differentially affected the content of different types of SCFAs in feces. The result indicated that only the GPTL group had a positive effect on microbial activities connected with SCFA production, whereas the GPTM and GPTH groups rather decreased SCFA concentrations. Ganpu tea at the lowest concentration showed positive effects on the concentrations of SCFAs such as acetic acid and propionic acid, whereas the concentration of butyric acid decreased. For branched short-chain fatty acids (BSCFAs) such as isobutyric acid, isovaleric acid, etc., Ganpu tea reduced their concentrations. The results of this study provide some references for further research on the health function of Ganpu tea. In addition, our study showed the feasibility of using Ganpu tea as a dietary therapy, but more research is needed before this dietary therapy is introduced.

## Figures and Tables

**Figure 1 nutrients-13-03715-f001:**
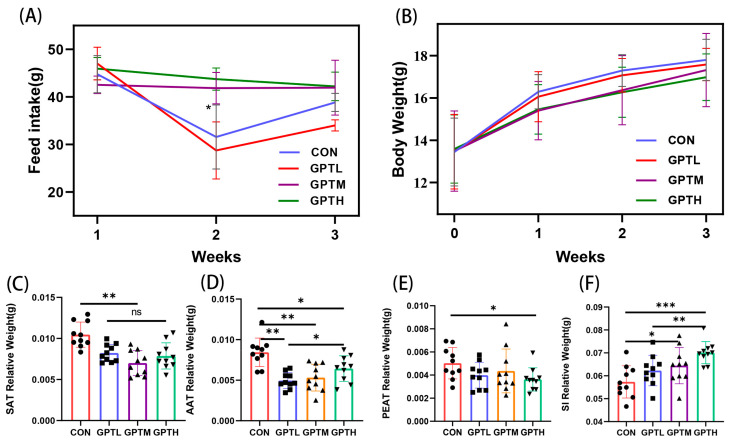
(**A**). Feed intake; (**B**). Body weight in mice for three weeks. (**C**–**F**): Effects of oral administration of Control group (CON), Low Concentration Ganpu tea group (GPTL), Medium Concentration Ganpu tea group (GPTM) and High Concentration Ganpu tea group (GPTH) on the relative weight of adipose tissue and small intestine to body weight in mice (*n* = 10, compared with each other * *p* < 0.05, ** *p* < 0.01, *** *p* < 0.001).

**Figure 2 nutrients-13-03715-f002:**
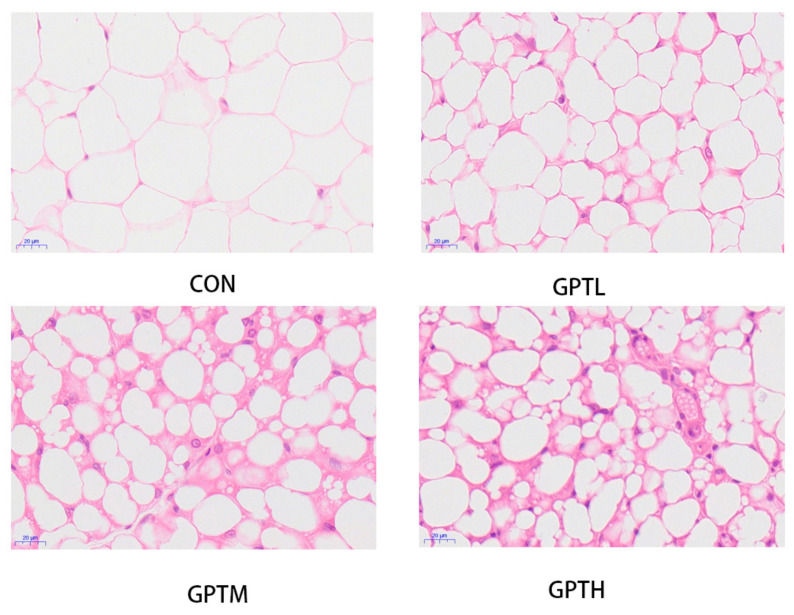
Hematoxylin and eosin (H&E) staining and morphology of stained abdominal adipose tissue (400×).

**Figure 3 nutrients-13-03715-f003:**
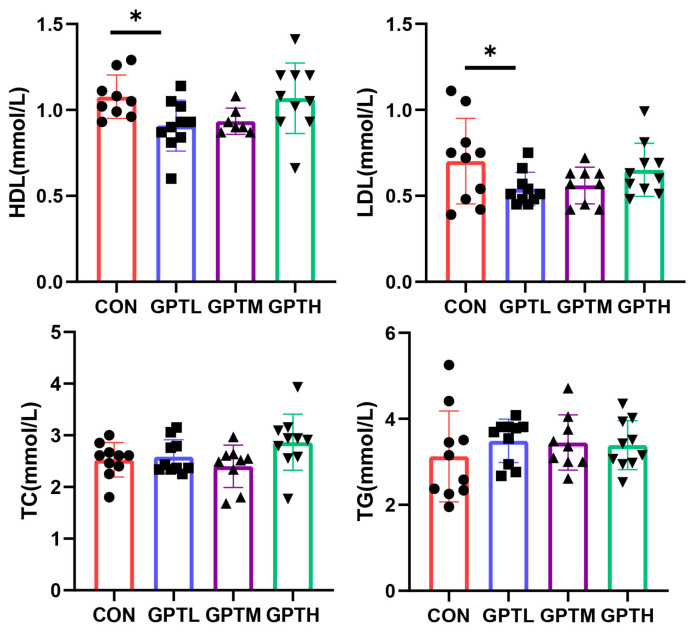
Serum lipid indices (* *p* < 0.05).

**Figure 4 nutrients-13-03715-f004:**
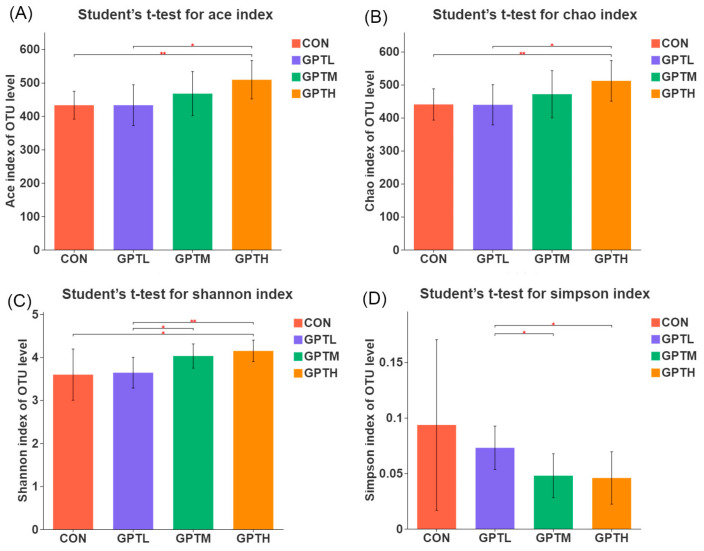
Effects of fed tea treatment on the α-diversity of the fecal microbiota in mice (*n* = 10). (**A**) ACE index, (**B**) Chao index, (**C**) Shannon index and (**D**) Simpson index of each group (*n* = 10, * *p* < 0.05, ** *p* < 0.01).

**Figure 5 nutrients-13-03715-f005:**
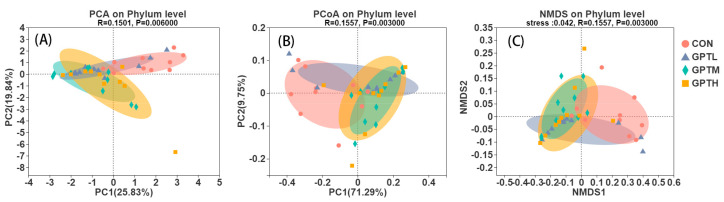
PCA(**A**), PcoA (**B**) and NMDS (**C**) plots of microbial communities were based on phylum level; each treatment group is represented by a different color (*n* = 10).

**Figure 6 nutrients-13-03715-f006:**
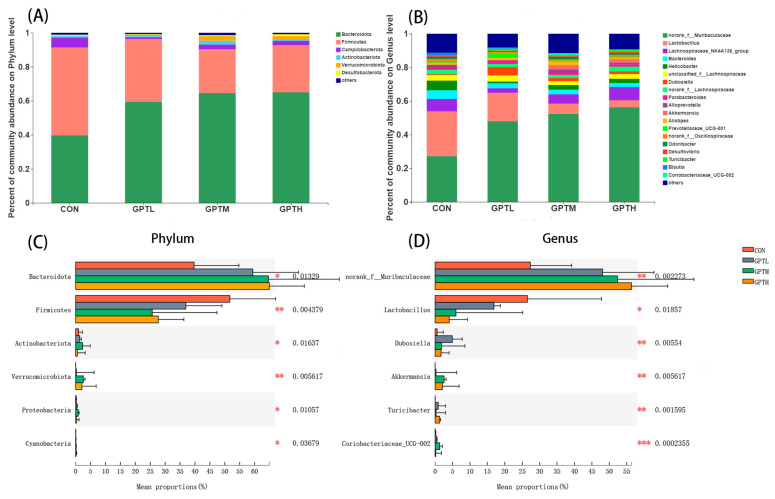
(**A**) Compositional change at the phylum level; (**B**) compositional change at the genus level; (**C**) analysis of species differences between groups (phylum level); (**D**) analysis of species differences between groups (genus level, * *p* < 0.05, ** *p* < 0.01, *** *p* < 0.001).

**Figure 7 nutrients-13-03715-f007:**
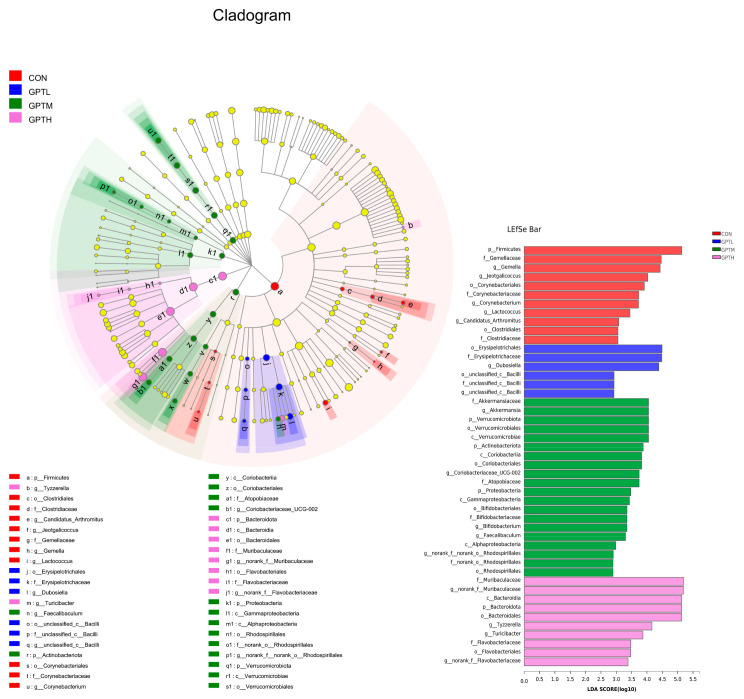
Differences in the gut microbiota between groups using linear discriminant analysis effect size (LEfSe) analysis from phylum level to genus level (*n* = 10). For taxa, which were defined as unclassified, no rank, ncultured or Incertae-Sedis, the name of a higher taxon level was added before its taxon abbreviation (p, phylum; c, class; o, order; f, family; g, genus; s, species).

**Figure 8 nutrients-13-03715-f008:**
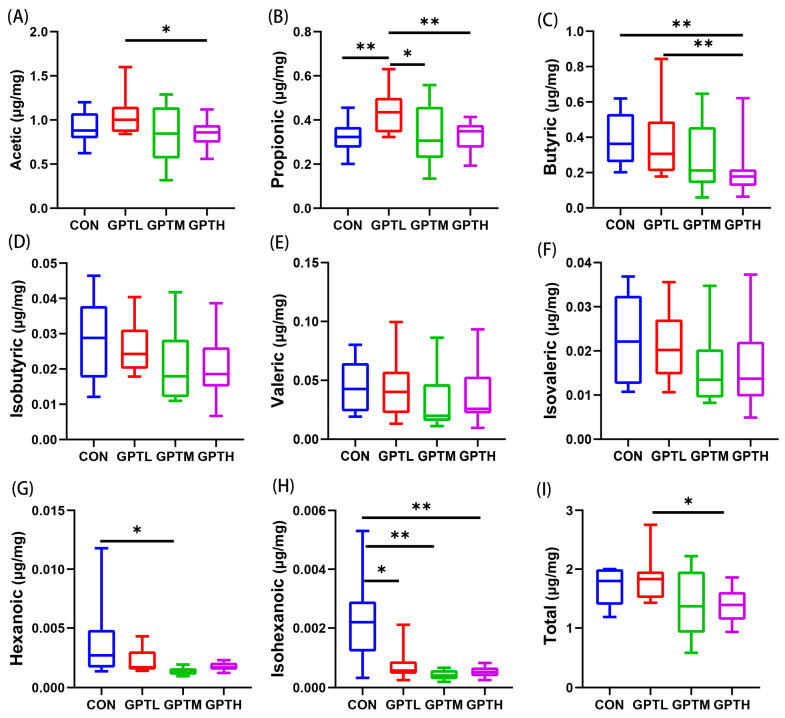
The changes of SCFAs in the fecal samples of the four groups. (**A**) Acetic acid, (**B**) propionic acid, (**C**) Butyric acid, (**D**) Isobutyric acid, (**E**) Valeric acid, (**F**) Isovaleric acid, (**G**) Hexanoic acid, (**H**) Isohexanoic acid and (**I**) Total SCFA concentration. * *p* < 0.05, ** *p* < 0.01.

**Figure 9 nutrients-13-03715-f009:**
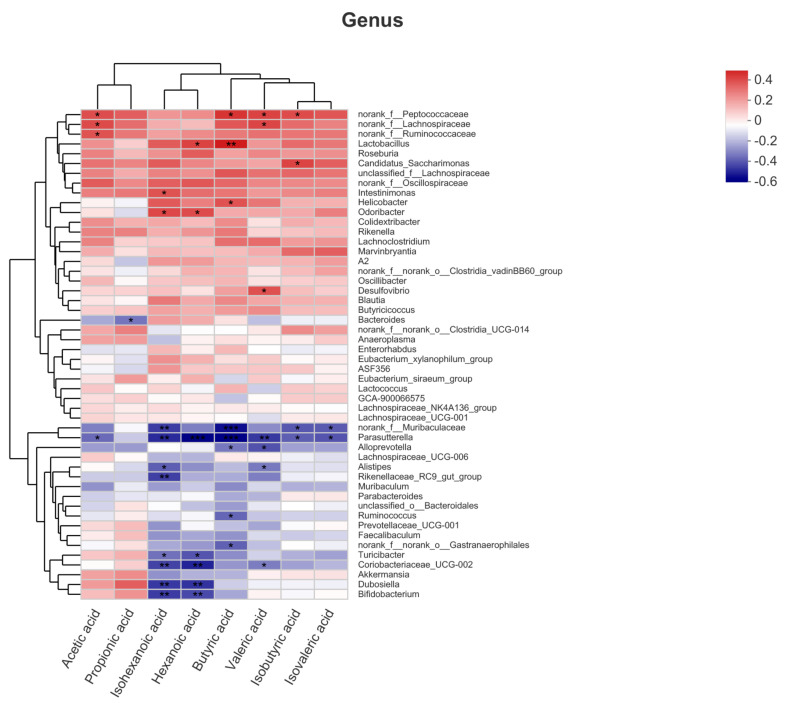
The Spearman correlation analysis between fecal microorganisms (genus level) and SCFAs. (* *p* < 0.05, ** *p* < 0.01, *** *p* < 0.001).

## Data Availability

16S rRNA data has been uploaded to the database, but not released. The SRA records will be accessible with the following link after the indicated release date: https://www.ncbi.nlm.nih.gov/sra/PRJNA729594.

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
