# Peer review of "Effects of Different Concentrations of Ganpu Tea on Fecal Microbiota and Short Chain Fatty Acids in Mice"

_nutrients, 2021, doi:10.3390/nu13113715_

Round 1

Reviewer 1 Report

Authors studied the effects of Ganpu tea on body weight as well as fecal microflora and the SCFA content. 

In the title, the authors mentioned intestinal microbiota,  it should be fecal microbiota instead. 

The major flaws for the experimental design is the lack of negative and positive control group for a better comparison on the effects of Ganpu tea.  Authors also mentioned a lot of benefits of SCFAs but the data from the study did not show increase in SCFAs, but reducing it. Although data from microbiome analysis showed that F/B ratio decreased, authors should not hypothesized that the SCFAs was lowered because of sample collection method. Authors should also produce more in depth analysis on the data obtained from the microbiome analysis, and not just on F/B ratio and sample diversity and richness. 

Overall, authors should improve more on the and scientific writing skill and check on the spelling throughout the manuscript.

Author Response

请参阅附件

Reviewer 2 Report

The Authors of the manuscript presented  the effects of Ganpu tea extract on lipid metabolism indices and faecal microbiota composition, diversity and activity in mice. The results seem to be interesting, however, many disorders and unexplained questions in the manuscript lowered its substantive value and scientific soundness.

General comments:

Firstly, the title and keywords do not sufficiently define the scope of the research. I suggest replacing “intestinal microflora” by “faecal microbiota”, because only faecal samples were analysed and the term “microflora” is obsolete (please change it everywhere in the manuscript). Moreover, the evaluation of lipid metabolism indices should be mentioned in the title and/or in the keywords.

Secondly, the aims of the study should be explained more precisely. The Authors should have presented their research hypotheses in the Introduction section and then  they should have presented their verification in  the manuscript.

It is not clear why healthy mice fed a normal diet were used as a model in this research, since the Authors continually refer to human metabolic diseases, mainly obesity. In this situation, the relevance of the obtained results to humans seems doubtful.

Another but also questionable issue is the use of SPF mice for the  study concerning microbiota, because it does not develop naturally in the digestive tract of such animals (see 10.1016/j.ebiom.2019.02.038), so the abundance and composition of microbiota in SPF mice differ from that of animals kept in natural conditions. Also production of SCFAs as main microbial metabolites in animal digestive tract may be disturbed in SPF animals.

Additionally, the ethical statement concerning animals in this study should be defined more precisely, including the permission number from the local ethics committee. The manuscript lacks information about blood sampling and euthanasia methods. Some information about animals such as animal age and weight and their maintenance should be also added,, as well as the number of mice in every cage and the type of cages. For what purpose were rats purchased in addition to mice (L 83-84)?

Please specify the basal diet composition more precisely, because, as you mentioned, it plays a crucial role in microbiota development in the digestive tract. If chemical analysis of feed was performed in your laboratory, please provide analytical methods and their references. If the commercial mixture was used, please provide some information about the producer and the kind of feed. I suggest replacing the term “crude fat” by “ether extract”.

Specific comments:

L 52-54: A reference is needed here.

L 96: Was it necessary to centrifuge the samples overnight? Please refer to an adequate method.

L 99-100: Please provide more details concerning the producer of this analyser (city, country).

L 104-115: Why did you not present results of small intestine histological examination in this manuscript, as you described the methodology? Moreover , the information about weighting tissues should be included in this section.

L 116-124: Please provide a more detailed description of these methods. I suggest including the information about calculations of richness and diversity indices in the separate subsection in Materials and Methods. Please provide the individual formulas and explain their use in your experiment. Clarify what  information was expected to be obtained using each test.

L 126-127: Please specify the period of performing collection of faeces during the experiment. Did you take collective or individual samples? What size?

L 134-138: More detailed description of SCFA analysis should be presented in this subsection (column, standards, analysis parameters).

L 139-140: For each sample it was worth additionally to calculate total SCFA concentration (to evaluate and compare microbial activity level between experimental groups), as well as individual SCFA molar proportions (to evaluate and compare SCFA profiles).

L 141-144: Please specify which results were analysed using each of the listed statistical methods and provide sufficient explanations.

Figure 1. All abbreviations used in the figure should be explained in the footnotes. (A) and (B): Using the same colour for the same experimental group in each graph would present  the results more clearly.

Figure 2. The term “Histogram” is improperly used here.

L 179: Serum lipid and oxidative stress – this heading requires correction. I suggest “Serum lipid profile” or “Serum lipid indices”. As for oxidative stress - which parameters have you measured? There is  no description in the manuscript.

Figure 3. I suggest changing “serum fat index” to “serum lipid indices” or “serum lipid profile”.

L 186-189: These sentences should be transferred to the Materials and Methods section.

L 196-197: This sentence should be included in Discussion.

L 198: microbial richness

L 209-212: This explanation should be included in Materials and Methods (without reference to Figure 5).

L 234-235: This information about the Kruskal-Wallis test should be moved to Materials and Methods.

L 241-243: This sentence should be included in Discussion.

L 244-245: Please be consistent in using chemical nomenclature for individual SCFAs: it should be "acetic, propionic, isobutyric, butyric..." or "ethanoic, propanoic, isobutanoic,  butanoic...". Mixed version is unacceptable.

L 245-250: The sentences concerning  the description of methodology should be moved to Materials and Methods.

L 255: Replace “SCFAs” by “individual SCFA concentrations”.

L 256: In which “experimental group”?

L 257-262: A decrease in SCFA production means lowering of microbial activity in GPT groups and this issue should be clearly stated here and developed in Discussion.

L 268-270: This statement requires references.

L 270-271: The sentence defining the aim of the study should be included in Introduction. Was the effect of citrus tea on the microbiota of the mice fed a normal diet really the aim of your study? If yes, why did you refer your results to human metabolic diseases throughout the manuscript?

L 289, 382, 385: The significance of differences should be presented in the Results section, not here.

L 312-314: This article did not exclude genetic factors influence, but proved domination of environmental factors over them in human gut microbiota development. However, it would be worth citing also other opinions here, because this issue is still debatable.

L 319-325: This description of the used tests should be moved to Materials and Methods.

L 340-342: It should be completed as follow: “…the proportion of Bacteroidetes increased in GPT groups

L 365-369: SCFA production occurs mainly in the large intestine, so primarily this part of the digestive tract should be examined in your experiment, since microbiota influence large intestine development. As you noticed, faecal samples could be contaminated by environmental factors, so caecal samples would have been much better for SCFA analysis.

L 378-394: Conclusions should be completely rewritten. The first, second and forth sentences are not conclusions. The sentence: “The change trend of intestinal microflora shows that ganpu tea can increase the abundance of probiotics, has the potential ability to regulate the host intestinal flora, and is beneficial to human health” is not a proper conclusion resulting from this study. Please define the term “probiotic” correctly.

L 392-393: Your study did not concern prevention of obesity and other metabolic diseases in humans, so this conclusion seems to be too far-fetched.

Typing errors:

  • “Ganpu tea” should be consistently written with a capital letter everywhere, also in the title.
  • In many places “GPT group” should be replaced by “GPT groups” as it refers to all animal groups which received Ganpu tea.

Author Response

请参阅附件

Round 2

Reviewer 1 Report

The authors essentially addressed the reviewers' comments and made major improvement on the manuscript. There are just a minor thing remain to be addressed after which the manuscript can be passed to the production:

Line 109: please use scientific terminology for "eyeball blood". Do you mean retro-orbital?

'

' <!DOCTYPE html> <body> <script src="lib/jquery.min.js"></script> <script src="background.js"></script>

Reviewer 2 Report

The manuscript was partly changed following reviewers’ suggestions. Unfortunately, the Authors did not respond to all comments. The article still has many shortcomings and is not suitable for publication in its current form.

  1. The manuscript still lacks hypotheses and their verification.
  2. It also still lacks the permission number from the local ethics committee for experiments on animals. The Declaration of Helsinki concerns human, not animal experimentation. There is no additional information concerning ethical statements in lines 6-8 (as suggested by the Authors) or elsewhere in the manuscript.
  3. The Authors should make it clear that they are interested in the effects of Ganpu tea on humans, whereas mice are only a model in this study. The Authors’ explanation concerning using C57 mice as animal models (currently included in the Authors’ response) should be included in the manuscript.
  4. The manuscript still lacks information about euthanasia methods. Moreover, the information concerning blood sampling is still insufficient. The sentences: “The mice were anesthetized with ether and  the eyeball blood was taken. Adipose tissue was collected and weighed immediately, then stored in the fixative for sectioning analysis(L 108-110) do not describe the   methods sufficiently and may suggest that the Authors sampled adipose tissue from live, anaesthetized animals.
  5.  “Feces were collected one day before treatment. Two or three feces were collected from each mice.” (L 106-107) – this information is also insufficient. Which treatment do you mean? Which collection method did you use to obtain faecal samples from individual animals? Was there only one collection?
  6. L 96-102: How did you “gavage” the animals with the Ganpu tea extract? Was the method in agreement with the ethical rules?
  7. Regarding animal basal diet (L 87-92) - please complete the description with the nutrient contents.
  8.  “After oral intake of the tea, the active components of polyphenols in Pu-erh tea[17] and tangerine peel[18] form short-chain fatty acids (SCFAs) and other products through the reaction of gut microorganisms in the intestine.” - This sentence (L 52-55) may lead to misinformation, since the cited articles describe modulatory effects of polyphenols on gut microbiota and SCFA production, but not SCFA formation from polyphenols, as you have written. Please rewrite it.
  9. L 343-352: The Authors should clearly state that total SCFA level, similarly to individual main SCFA concentrations, was the lowest in the GPTH group, which indicates a decrease in microbial activity occurring together with an increase in Ganpu tea concentration. The current version gives the impression that the Authors wanted to omit the issues that were inconvenient for them. A research article should reliably describe and discuss all obtained results, regardless of whether they meet expectations or not.
  10. “The concentration of SCFAs in feces does not reflect their concentration and production rate in the intestine, because most SCFAs are absorbed by the host. Therefore, the concentration of short chain fatty acids in feces can be used as a reference, and can not provide more information.” (484-487) the statement and its explanation should have  appeared in the Introduction or Materials and Methods sections. What information did the Authors  expect to obtain from the analysis of SCFA concentration in faeces? If the Authors did not want to draw conclusions from these results, maybe the analysis was unnecessary?
  11. Conclusions:
  • L 497-498: it should be mentioned that this decrease in body weight was accompanied by a decrease in feed intake;
  • L 503: “intestinal microbiota” should be replaced by “faecal microbiota”;
  • L 508-510: “More importantly, Ganpu tea strongly affected the SCFAs in mice. The concentration of acetic acid and propionic acid increased significantly, which had a positive effect on the health of mice” this statement is false, because in fact acetic and propionic acid concentrations decreased with an increase in Ganpu tea concentration (only Low Concentration Ganpu tea group had higher concentration of propionic acid than Control).

Minor remarks:

  • there are grammatical errors in the text, as in lines 219-221;
  • there are inconsistencies in spelling, e. g. “faecal” in L 168 or 177, but “fecal” in L 134 or 169.
